# Gut Microbiota Modulation of Short Bowel Syndrome and the Gut–Brain Axis

**DOI:** 10.3390/nu15112581

**Published:** 2023-05-31

**Authors:** Jasmine Carter, Jeffery Bettag, Sylvia Morfin, Chandrashekhara Manithody, Aakash Nagarapu, Aditya Jain, Hala Nazzal, Sai Prem, Meghan Unes, Matthew McHale, Chien-Jung Lin, Chelsea Hutchinson, Grace Trello, Arti Jain, Edward Portz, Arun Verma, Marzena Swiderska-Syn, Daniel Goldenberg, Kento Kurashima

**Affiliations:** Department of Pediatrics, Saint Louis University, Saint Louis, MO 63104, USA

**Keywords:** short bowel syndrome, gut microbiome, dysbiosis, gut microbiota, gut-brain axis, gastrointestinal disease, probiotics

## Abstract

Short bowel syndrome (SBS) is a condition that results from a reduction in the length of the intestine or its functional capacity. SBS patients can have significant side effects and complications, the etiology of which remains ill-defined. Thus, facilitating intestinal adaptation in SBS remains a major research focus. Emerging data supports the role of the gut microbiome in modulating disease progression. There has been ongoing debate on defining a “healthy” gut microbiome, which has led to many studies analyzing the bacterial composition and shifts that occur in gastrointestinal disease states such as SBS and the resulting systemic effects. In SBS, it has also been found that microbial shifts are highly variable and dependent on many factors, including the anatomical location of bowel resection, length, and structure of the remnant bowel, as well as associated small intestinal bacterial overgrowth (SIBO). Recent data also notes a bidirectional communication that occurs between enteric and central nervous systems called the gut–brain axis (GBA), which is regulated by the gut microbes. Ultimately, the role of the microbiome in disease states such as SBS have many clinical implications and warrant further investigation. The focus of this review is to characterize the role of the gut microbiota in short bowel syndrome and its impact on the GBA, as well as the therapeutic potential of altering the microbiome.

## 1. Introduction

The average length of an adult’s small intestine is typically around 600 cm [1,2,3]. In infants, the length of the small intestine is more difficult to determine, as the small intestine is actively growing and developing. Many factors can change the efficiency of intestinal absorption, such as the overall physical length of the bowel and the composition of the bacterial population [4,5]. The condition that inherently reduces the length of the small intestine, either congenitally or because of an iatrogenic intervention, is termed short bowel syndrome (SBS). SBS causes malabsorption as the majority of the nutrients obtained from an oral diet are absorbed in the small intestine. Although quite often secondary to surgical resection, SBS can be a result of several diseases, injuries, or conditions that disturb the normal function of the small intestine.

In pediatrics, the definition of SBS is more specific, with a requirement for less than 25% of remaining small bowel for gestational age [6,7]. The most common cause of SBS in infants is necrotizing enterocolitis (NEC), which is characterized by necrosis of the bowel wall with subsequent perforation [3,8,9]. The cause of NEC is often unknown, but typically occurs in premature infants or infants with underlying conditions such as neonatal respiratory distress syndrome (NRDS) or in formula-fed infants with immature immune systems. Infants can also be born with congenital short bowel syndrome due to dysgenesis. The exact prevalence of SBS is unknown, with little data on racial or gender bias. In the United States, there are approximately 20,000 reported cases [10,11]. The prevalence of SBS in children has been reported to occur in 24.5 out of 100,000 live births [12,13,14].

Recognizing the importance of a healthy gut microbiome is a growing field of research. As more data becomes available, its role in host growth and development and relationship to pathology is becoming more important. Studies are beginning to analyze the role of microbiota in human homeostasis and disease as a result of the microbiota composition [15]. Recent data have focused on the role of the microbiome in disease processes such as SBS [16,17]. Just as nutritional balance is altered in the state of SBS due to a lack of intestinal tissue, the microbiome changes of different microenvironments are lost as well. This process is known as intestinal adaptation. Achieving adaptation following a shortened bowel is key to reducing morbidity in such patients [18].

The focus of this review is to characterize the role of the gut microbiota in both positively and negatively modulating short bowel syndrome, and if altering the microbiome with factors such as pre-, pro-, or symbiotics has therapeutic potential.

## 2. Gut Microbiota

The human microbiome is composed of bacteria, archaea, viruses, and microbes. Healthy adult humans typically have over 1000 species of bacteria; however, most of these species belong to just a few phyla [19]. In adults, Bacteroidetes and Firmicutes are the dominant phyla found in the gut [20], while other phyla such as Actinobacteria, Proteobacteria, and Verrucomicrobia are present in smaller numbers [19]. Gut bacteria have a multitude of functions that are still being discovered. Some known functions include digestion, vitamin production, regulation of immune response via metabolites, lipid metabolism, and protection against pathogens [21]. Gut microbiota is impacted by a variety of factors such as medications, environment, and genetics, but is most importantly affected by diet. It has been found that food components, particularly carbohydrates, can ferment and result in the production of short-chain fatty acids (SCFAs), branched-chain fatty acids, ammonia, amines, phenolic compounds, and gases. In recent years, SCFA has been identified as a link between host nutrition and maintaining homeostasis in the intestine. SCFAs are used as fuel for intestinal epithelial cells (IEC) and for IEC regulation to aid in cell proliferation and differentiation [22]. A recent study, which analyzed the intestinal microbiome, SCFAs, and intestinal adaptation in a probiotic and antibiotic treatment of neonatal piglets with SBS, concluded that antibiotic use was associated with a decrease in microbial diversity and an increased abundance of proteobacteria, and the probiotic group noted an increase in microbial diversity and SCFA composition [23,24].

However, attempts to characterize the microbiome of healthy individuals have been difficult [19]. Many projects have been implemented, such as the MetaHIT and Human Microbiome Project (HMP) to collect data on the microbiome of individuals globally using 16S rRNA-encoding gene sequencing [20]. During the early stages of these projects, it was believed there were “core microbiota” in all healthy individuals that were absent or scarce in diseased states [25], however, recent evidence leads away from this hypothesis [26,27]. What is becoming more apparent, is that the gut microbiome is dependent on factors that had previously not been considered, including age, geographical location, history of disease, etc. A study by Caporaso et al. performed a time series analysis of the human microbiome, which concluded that there is variability in each person’s microbiota across days, weeks, and months. They also noted that only a small amount of the total taxa was consistent, and this agrees with recent data that suggests there are no “core” bacteria that represent a standard healthy individual [28]. Furthermore, when metagenomic sequencing was performed on the gut microbiome of 34,057 individuals from the United States and Israel, it was noted that traits such as age, glycated hemoglobin (HbA1C%), and BMI could be predicted by linear models and decision tree models based on an algorithm created to analyze bacterial relative abundance [29].

Indeed, it has been shown that there is typically a larger difference in gut microbiota in industrialized countries versus those in developing countries. In order to verify this hypothesis, fecal microbiota of US residents was compared to that of natives in rural Papua New Guinea. It was noted that the rural natives of Papua New Guinea had greater microbial diversity than that of the US residents. These findings suggest that the alteration in the microbiome of more industrialized societies may come from lifestyle factors, which include an increase in processed foods, as well as differences in fiber, fruits, and vegetable consumption [30,31] that limit bacterial diversity [32,33]. Additional studies analyzed the fecal microbiota in European children and that of children from a rural village in Africa, Burkina Faso [34], which showed a significant difference in the microbiome between the two groups. The children from Burkina Faso had a microbiota rich in Bacteroidetes and lower in Firmicutes [34]. Additionally, the children of Burkino Faso had an abundance of bacteria from the genus Xylanibacter and Prevotella—which was completely absent in the European cohort—as well as significantly more SCFAs, as compared to the European children [34]. It was hypothesized that the gut microbiome of the African cohort coevolved with their high-fiber diet, which led to improved health and allowed them to receive maximum energy while remaining protected from colonic inflammation [34]. 

## 3. Dysbiosis in SBS

While it is still unknown what a “healthy” microbiome constitutes, there are many studies that aim to characterize a “bad” microbiome. Dysbiosis is the term that is used to refer to a disturbance in gut microbiota homeostasis caused by an imbalance in the gut flora, alterations in functionality and metabolic changes, and/or changes in distribution. However, just as it has been difficult to determine a healthy microbiome, defining dysbiosis and its cause(s) has been equally challenging. There has been conflicting data on distinguishing between the cause and effect of dysbiosis. Does disease pathogenesis cause dysbiosis, or does dysbiosis cause disease? Many diseases have been associated with dysbiosis, including inflammatory bowel diseases (IBD), obesity, allergies, colorectal cancer, and cognitive development [35]. Indeed, research has identified three main pathogens that have been associated with IBD: Mycobacterium avium paratuberculosis, adherent-invasive Escherichia coli, and Clostridium difficile [36]. Recent data suggests both brain–gut and gut–brain dysfunctions occur in irritable bowel syndrome (IBS), a specific type of bowel disease [36]. Disruption of the gut–brain axis determines changes in intestinal motility and secretion, as well as alterations of the enteroendocrine and immune systems [36].

The gut microbiota has also been found to be linked to several metabolic disorders, such as diabetes and obesity. However, these links have been made using small cohorts of individuals and are not consistent across all studies [29]. Some research has shown that having a diverse microbiome is a positive indicator of health, while others have noted that this diversity is associated with microbiome instability [29].

When dysbiosis occurs in SBS, it typically results in small intestinal bacterial overgrowth (SIBO) [7,37]. SIBO, in the small bowel, is defined as having >105 CFU/mL of bacteria or >103 CFU/mL of bacterial species that are typically found in the colon [37]. SIBO can have a variable presentation including feeding intolerance, bloating, and abdominal pain and thus ameliorative strategies remain a focus in treating patients suffering from SIBO [38,39].

To make this diagnosis, an aspirate from the small bowel is required, which is typically difficult to obtain in pediatric patients and usually requires an endoscopy [37]. An alternative way to make this diagnosis is by performing a hydrogen breath test in which the patient ingests a carbohydrate, and then the amount of H2 exhaled is measured. More commonly, SIBO is diagnosed based on clinical symptoms that typically improve with antibiotics [37]. However, another study concluded that rather than a bacterial overgrowth in SBS, it seems that dysbiosis is more related to an imbalance in the prominent phyla in the small intestine [40,41].

It has also been hypothesized that alterations in the gut microbiota enable bacterial flux across the intestinal mucosa which results in cytokine mediated hepatocellular injury [42,43,44]. Notably, significantly elevated Interleukin-6 (IL-6) and tumor necrosis factor-α (TNF-α) have been reported in SBS [45,46,47]. Upon initiation of enterally administered antibiotics, a reduction in liver injury has been reported [48,49,50].

Additionally, there is emerging evidence that the gut microbiome communicates with the central nervous system vagally via the gut–brain axis, which then modulates brain and behavioral development [51].

## 4. The Gut–Brain Axis

The gut–brain axis (GBA) is made up of bidirectional communication between the enteric and central nervous systems (CNS). This bidirectional network includes the autonomic nervous system (ANS), CNS, the brain and spinal cord, enteric nervous system (ENS), and the hypothalamic pituitary adrenal (HPA) axis. The autonomic system drives afferent signals, which arise from the lumen and are transmitted through neuronal connections to the CNS, and then from the CNS to the intestinal wall. The core stress–efferent axis is the HPA axis, which coordinates the responses of the organism to stressors. The HPA axis is part of the limbic system of the brain, predominantly involved in emotional responses and memory. Multiple studies have demonstrated that stressful stimuli have an effect that results in increased permeability of intestinal epithelium [52], which allows bacterial antigens to penetrate the gut epithelium [52], driving immune responses in the intestinal mucosa. An increase in permeability can also drive endotoxin mediated hepatic and systemic inflammatory responses [53,54]. Stress-induced changes in intestinal mucosa permeability involve activation of specific cells in the intestine (glial and mast cells), overproduction of interferon-γ, and morphological changes in colonic epithelium. These are postulated to occur as a result of reduced expression of tight junction protein 2 (zona occludens 2) and by components of the intestinal tight junction modulation [52]. Environmental stresses activate the HPA axis by releasing corticotropin-releasing factor (CRF) from the hypothalamus and stimulating the secretion of adrenocorticotropic hormone (ACTH) from the pituitary gland leading to the release of cortisol, a major stress hormone, from the adrenal glands, (Figure 1). It can thus be concluded that both neural and hormonal lines of communication allow the brain to influence the activities of intestinal functional effector cells. The function of these cells is also regulated by the gut microbiota, whose role in the crosstalk between the gut and brain has been a major research focus.

### 4.1. Gut Microbiome and Its Role in the Gut–Brain Axis

There is evidence both clinically and experimentally that suggests enteric microbiota has an important impact on GBA (Figure 2). Microbiota has been found to interact with local intestinal cells, the ENS, and the CNS via several metabolic and neuroendocrine pathways. Evidence of this interaction was first found in a study which showed significant improvements in patients with hepatic encephalopathy after they were treated with oral antibiotics [36]. More recently, there has been emerging evidence supporting the role of microbiota and its influence on cognition, brain structure, performance, neurological disorders, and mental disorders such as anxiety and depression [55,56].

A study investigating the behavior of germ-free mice after being placed in a maze supports this claim. In comparison to specific pathogen-free (SPF) mice, it was noted that germ-free mice exhibited basal behaviors that could be interpreted “anxiolytic” compared to the SPF mice, supporting the idea that microbiota influences anxious behaviors [57]. Another experiment hypothesized that gut microbiota responds to external environmental signals that regulate brain development and function and it was concluded that altered expression of canonical signaling pathways, synaptic-related proteins, and neurotransmitter turnover may contribute to behavioral differences observed between SPF and GF mice [58]. Emerging data supports that alteration in tryptophan metabolism as influenced by the gut microbiota could drive key signaling along the GBA [41,59,60].

What seems apparent is that the link between the gut–brain axis and the role it plays in gastrointestinal orders such as SBS remains an area ripe for research. Thus far research focusing on the gut–brain axis and other gastrointestinal disorders such as IBD [61,62] has emphasized the bidirectional communication between the gut and the nervous system. From this, we can conclude that the brain may influence the microbiota in individuals, but the exact nature of that link is currently unknown.

### 4.2. Short Bowel Syndrome and Gut Microbiome

As previously noted, changes in microbiota can be adaptive and respond positively to assist in disease states such as SBS and lack of luminal nutrition or can be pathologic as a result of maladaptive change [17,63,64]. One question that remains key is the causal relationship between SBS and microbiota. Do bacteria in SBS become maladaptive as a result of the disease, or do previously dysregulated microbiota increase susceptibility to conditions that may result in SBS? One of the more commonly reported associations between SBS and gut bacteria is in the context of clinical expression of metabolic acidosis. It is well documented that SBS results in increased lactic acid due to increased delivery of carbohydrates to the colon [65]. Indeed, a study found a 10-fold increase in D-lactate in the colon of SBS vs. control patients. Analyses revealed that total fecal bacteroidetes significantly decreased, however the ratio of lactobacilli to other bacteria was elevated, particularly of lactobacilli that produce D-lactate [66]. In these instances, the alterations in microbiota are likely secondary to the state that SBS induces and result in clinically significant metabolic acidosis. However, some studies pose the increase in lactobacilli as adaptive, but with the negative side effect of acidosis. Sommovilla et al. published a murine study with small bowel resection and the alteration in microflora. In healthy control versus surgically resected mice, no difference was found in microbiome diversity or in the long-term outcomes of intestinal flora; however, there was a short-term overall decrease in bacteroidetes and proportional increases in lactobacilli. It was thus posed as a temporary adaptive response to increase energy absorption in the colon [67]. Further studies have verified the increase in lactobacilli across adult and pediatric patients [68,69].

The microflora is further altered in the context of antibiotic usage in SBS patients. Because of the risk of SIBO, infants are prophylactically placed on medications such as metronidazole or bactrim. In a study on pediatric SBS, while healthy controls had 54% bacteroidetes and 33% firmicutes, SBS patients without antibiotics had 49% firmicutes, and SBS patients with antibiotics had proteobacteria clonal proliferation at 56–58% [70]. Given the variability of the gut microbiome in the different regions of the small bowel, the site and extent of bowel resection in SBS as well as antibiotic usage influences key patient outcomes such as time to enteral autonomy, growth, and risk of SIBO [71,72].

Interestingly, studies have also drawn comparisons to the microbiome in IBD such as Crohn’s or ulcerative colitis (UC) and that of SBS. It is believed that within the intestinal mucosa, chronic, subtle, and low-grade inflammation, driven by alterations in the gut microbiome may mediate the phenotypic characteristics in IBD and SBS [73]. In a study in pediatric patients, it was noted that the alterations in microbial diversity of SBS patients were consistent with the intestinal mapping of Crohn’s and UC, and a majority of SBS patients on TPN had these changes. However, no significant difference between SBS patients weaned from TPN and healthy controls was noted, indicating that the state of SBS, which necessitates TPN, may be more important than the lack of intestine itself [74]. Overall, evidence in the literature points to microbial dysregulation as a consequence rather than cause of SBS, with an emphasis on the concurrent state of nutrition, i.e., TPN vs. EN [75,76,77,78,79].

SBS can present in several variants based on anatomical location and length of resection or missing tissue [80] (Table 1). Additionally, SBS can result in an altered microbiome, resulting in SIBO or dysbiosis. In a study conducted by Budniska et al. in 2019, apart from shared characteristics and species between all categories, each type had a unique composition. The categories used in the study—SBS I (jejunostomy), SBS II (jejunocolonic), and non-PN SBS (jejunocolic without TPN dependence)—have been shown to have a different composition of microflora. Patients from each classification were analyzed and results showed that SBS I patients had a high load of aerobic bacteria, with concurrent depletion of anaerobes. This study also confirmed some normalization of the fecal microbiome in non-TPN SBS patients. It also described a unique difference. Non-TPN SBS had high phenol content, while both SBS I and II were enriched by chenodeoxycholic and deoxycholic acid while being depleted of lithocholic acid [81]. Additionally, data have also shown that resection length may affect the gut–brain axis. Pertinently, a study on SBS patients noted that a shorter bowel was linked to more proteobacteria, and longer remaining small bowel led to the presence of higher firmicutes, with a reference cutoff at 35 cm underscoring a correlation between composition of gut flora and type, size, or location of small bowel resection.

### 4.3. Therapeutic Approaches to the Microbiome

Probiotics are used to add healthy bacteria to the gut and could theoretically be used to restore the alterations in gut microbiome that are the consequence of SBS. It has been noted that probiotics may be helpful in recovering from acute diarrhea, as noted from a study investigating the effect of lactobacillus based probiotics on acute diarrhea in 71 well-nourished children between the ages of four to forty-five months [82]. Sets of participants received lactobacillus GG-fermented milk product twice daily (group 1), lactobacillus GG powder twice daily (group 2), or the placebo (group 3), which was pasteurized yogurt twice daily over five days, in addition to a normal diet. The average duration of diarrhea after the treatment was shorter in groups 1 and 2 vs. group 3 [82]. No significant alterations in intestinal permeability were noted, which indicated an absence of mucosal disruption, and thus was beneficial for recovery from diarrhea [82]. However, the previously mentioned study was conducted in otherwise healthy children, thus the results may not be representative of children suffering from SBS. Emerging data also supports the use of fecal microbiota transplant as a therapeutic strategy for SBS with varying success, though the data remains compelling [83,84]. Additionally, there is continued exploration of microbiome signatures as potential biomarkers of intestinal adaptation and as a driver of D-lactic acidosis [63,85,86].

Another study analyzed the effects of probiotic supplementation in children with SBS versus a healthy control group [87]. In a randomized control study, eighteen children with SBS after being weaned from TPN were randomly given probiotics (Lactobacillus rhamnosus and Lactobacillus johnsonii) or a placebo daily for two months [87]. It was found that the abundance of Lactobacillus did not change at baseline or at the end of the study between the two cohorts, thus concluding that Lactobacillus probiotics did not result in a change in the fecal microbiota or overall growth in comparison to the placebo [87]. Indeed, the use of probiotics as a way of restoring alterations of gut microbiome in those suffering from SBS warrants further investigation.

Gut microbiota and the role of diet was assessed in a study of children with intestinal failure. There was an inverse relationship between parenteral nutrition support and microbial diversity as well as SCFA profile. Higher enteral nutrition was associated with a microbiota function and structure closer to health controls. The route of administration of nutrition also drove a supraphysiological increase in harmful pathobionts, underscoring the importance of enteral nutrition in microbial restoration and intestinal adaptation in SBS [88].

## 5. Conclusions

SBS is a life-threatening condition associated with significant morbidity and mortality. Many effects are likely driven from alterations in the gut microbiome, which modulates key signaling pathways. Recent studies have shown that alterations in gut structure hold implications for the microbiome. In this review, we discussed the influence of the gut microbiome in SBS and other disease states. Furthermore, studies have shown microbial alterations are secondary to SBS rather than a predisposing factor. In many cases, investigation into the microbiome in SBS patients revealed a significant alteration in the composition of the microbiome.

Additionally, even minor alterations of the gut–brain axis, and signaling molecules such as SCFAs that are byproducts of bacterial metabolism have been shown to increase inflammatory states. These molecules may have implications in neurological development in infants. Additionally, not only have the composition and bacterial metabolites been shown to play a role in SBS, but the exact structure, characterization, and anatomical location of small bowel loss also hold profound implications for microbial richness and diversity. Given the diversity of species and genera of gut flora, there is a myriad of pathways, metabolites, and implications of these interactions that likely have significant effects and need further studies. Some studies have investigated probiotics as a therapeutic strategy, but few results suggest probiotics are therapeutic for those suffering from SBS. Uncovering the complex relationships between the gut microbiome, the GBA, and SBS remains a major focus and opens the potential for personalized medicine approaches based on individual gut microbiome profiles [89,90].

Ultimately, the role of the microbiome in disease states such as SBS have many clinical implications and warrant further investigation in both prophylactic and therapeutic settings.

## Figures and Tables

**Figure 1 nutrients-15-02581-f001:**
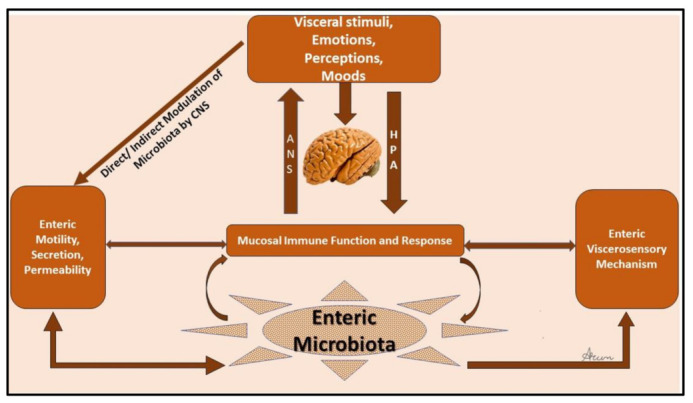
Schematic of the gut–brain axis.

**Figure 2 nutrients-15-02581-f002:**
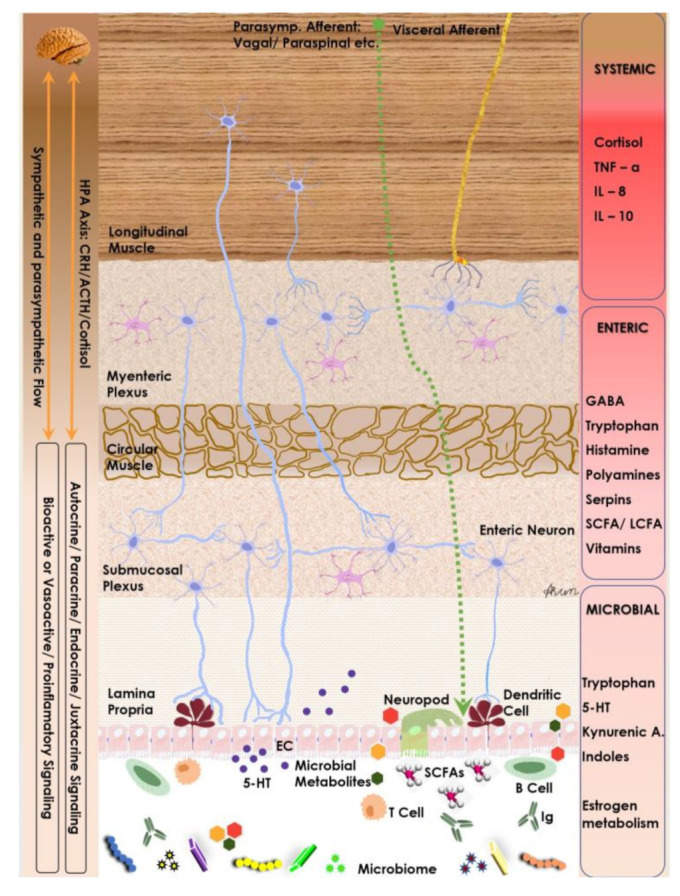
Neuro-endocrine and metabolic components of the gut–brain axis. Note the interplay between microbiome, its metabolites and host endocrine & immune cells as they modulate the enteric nervous system and are modulated by the host autonomic flow.

**Table 1 nutrients-15-02581-t001:** Classification of short bowel syndrome and probability of TPN dependence.

CATEGORY:	DESCRIPTION:	PROBABILITY OF TPN DEPENDENCE
**SBS JEJUNOSTOMY ANASTOMOSIS**	The colon, ileum, and part of the jejunum are removed. A stoma is created in the abdomen from the remaining jejunum.	Higher in patients with <150 cm of jejunum remaining—Variable
**SBS JEJUNOCOLONIC ANASTOMOSIS**	A portion of the ileum, along with (sometimes) a portion of the colon are resected, and the jejunum is re-connected to the colon.	Higher in patients with <60–65 cm jejunum remaining—Variable
**SBS JEJUNOILEAL ANASTOMOSIS**	A portion of the jejunum, along with part of ileum are resected, and then remaining sections are re-joined.	Higher in patients with <35 cm jejunum remaining—Low

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
