# Peer review of "Gut Microbiota Modulation of Short Bowel Syndrome and the Gut–Brain Axis"

_nutrients, 2023, doi:10.3390/nu15112581_

Round 1

Reviewer 1 Report

This is a very nice article about a passionate subject which needs further investigation in order to gather evidence to make recommendations of the use of probiotics in SBS.

I believe that you should add information about the potential usefullness of Fecal Microbiota transplantation which was reported as effective in 1 SBS case (Davidovics ZH, Vance K, Etienne N, Hyams JS. Fecal transplantation successfully treats recurrent D-lactic acidosis in a child with short bowel syndrome. JPEN J Parenter Enteral Nutr. 2017;41(5):896-897.)

And maybe also comment about the use of microbiome as a potential biomarker of intestinal adaptation and risk of D lactic acidosis. 

I suggest another paper  The Gut Microbiome in Patients with Intestinal Failure: Current Evidence and Implications for Clinical Practice Esther Neelis, MD1 ; Barbara de Koning, MD, PhD1; Edmond Rings, MD, PhD1,2; Rene ́ Wijnen, MD, PhD3; Ben Nichols, PhD4; Jessie Hulst, MD, PhD1;

and Konstantinos Gerasimidis, PhD4Journal of Parenteral and Enteral Nutrition

Volume 00 Number 0xxx 2018 1–12 

Reviewer 2 Report

The authors of this review article have undertaken a comprehensive examination of the role of the gut microbiome in Short Bowel Syndrome (SBS), a life-threatening condition associated with significant morbidity and mortality. By exploring the relationship between SBS and the gut microbiome, the authors have highlighted the importance of understanding the complex interactions between gut structure, microbial composition, and signaling pathways in the context of this disease.

The review discusses the influence of the gut microbiome in SBS and other disease states, with a focus on how microbial alterations are secondary to SBS rather than predisposing factors. The authors emphasize the significance of the gut-brain axis and the role of bacterial metabolites, such as short-chain fatty acids (SCFAs), in inflammatory states and neurological development in infants.

In addition to the composition and bacterial metabolites, the authors also discuss the importance of understanding the role of gut structure, characterization, and anatomical location of small bowel loss in determining microbial richness and diversity. This highlights the need for further research to uncover the myriad pathways, metabolites, and implications of these interactions.

The authors have undoubtedly conducted a thorough and comprehensive examination of the complex interplay between the gut microbiome and Short Bowel Syndrome (SBS). Their work contributes valuable insights into the potential therapeutic strategies and clinical implications of understanding the gut microbiome in the context of SBS. However, major revisions are necessary to address some of the limitations and gaps identified in the review.

Gut microbiota:

1.      In the section comparing gut microbiota in industrialized and developing countries, please provide more information on the lifestyle factors that may contribute to the observed differences in gut microbiota composition. Also, consider discussing the potential health implications of these differences.

2.      In the comparison of the gut microbiota of European children and children from Burkina Faso, consider elaborating on the possible consequences of their differing gut microbiota composition for their overall health and susceptibility to diseases.

3.      In the discussion of SIBO in SBS, please provide more information on the potential causes of SIBO in these patients and the implications of SIBO for their overall health and disease management.

Gut-brain

4.      When discussing the bidirectional communication between the enteric and central nervous systems, consider providing more details on the specific types of signaling molecules and pathways involved in this communication.

5.      Discuss the role of the gut microbiota in the Gut-Brain Axis, including the specific mechanisms through which the gut microbiota influences the communication between the gut and the brain.

6.      When discussing the effects of stress on intestinal permeability, consider providing more information on the potential consequences of increased intestinal permeability and how it may impact overall health and well-being.

Gut Microbiome and its role in the Gut-Brain Axis

7.      When discussing the influence of gut microbiota on neurological and mental disorders, provide a broader overview of the research findings in this area, including the potential mechanisms through which gut microbiota may contribute to these disorders.

8.      Explore the potential therapeutic implications of manipulating the gut microbiome to improve neurological and mental health, including the use of prebiotics, probiotics, and fecal microbiota transplantation.

Short Bowel Syndrome and Gut Microbiome

9.      When discussing the role of antibiotics in SBS patients, consider elaborating on the potential benefits and drawbacks of antibiotic treatment, including the impact on gut microbiome composition and the risk of antibiotic resistance.

10.   In the comparison of SBS and inflammatory bowel disease (IBD), provide a more detailed overview of the similarities and differences in gut microbiome composition and discuss the potential implications for the pathogenesis and treatment of these conditions.

11.   When discussing the impact of anatomical location and length of resection on the gut microbiome and gut-brain axis in SBS patients, consider elaborating on the potential mechanisms underlying these associations and their clinical significance.

Therapeutic Approaches to the Microbiome

12.   Expand your discussion of probiotics to include a wider range of potential benefits and risks in the context of SBS. Discuss the various factors that could influence the effectiveness of probiotics, such as strain selection, dosage, and duration of treatment.

13.   Consider discussing the use of prebiotics as an alternative or complementary approach to modulating the gut microbiome in SBS patients. Explain the potential benefits and drawbacks of this approach compared to probiotics.

14.   Address the potential role of fecal microbiota transplantation (FMT) in restoring gut microbiome alterations in SBS patients. Discuss the available evidence, potential risks, and future research directions for this approach.

15.   Discuss dietary interventions that could be employed in conjunction with microbiome-targeted therapies to improve outcomes in SBS patients. Consider the potential benefits and challenges associated with different dietary strategies, such as specific carbohydrate diets or fiber supplementation.

Conclusions:

16.   Emphasize the importance of future research in uncovering the complex relationships between the gut microbiome, the gut-brain axis, and SBS. Highlight the potential for personalized medicine approaches based on individual gut microbiome profiles.

Author Response

Response: Thank you.

Critique:

  1. In the section comparing gut microbiota in industrialized and developing countries, please provide more information on the lifestyle factors that may contribute to the

observed differences in gut microbiota composition. Also, consider discussing the potential health implications of these differences.

Response: Thank you for the critique. As per reviewer critique we expanded the section on lifestyle factors and have added 2 additional references.

Critique:

  1. In the comparison of the gut microbiota of European children and children from Burkina Faso, consider elaborating on the possible consequences of their differing gut microbiota

composition for their overall health and susceptibility to diseases.

Response: Thank you for the critique. While health outcomes were not the focus of this review, we have included the overall improved health as per reviewer critique. We can expand further should the reviewer / editorial committee desire.

Critique:

  1. In the discussion of SIBO in SBS, please provide more information on the potential causes of SIBO in these patients and the implications of SIBO for their overall health and disease management.

Response: Thank you for the critique. Implications of SIBO have been added. We also included 2 additional supporting references.

Critique:

  1. Gut Brain Axis: When discussing the bidirectional communication between the enteric and central nervous systems, consider providing more details on the specific types of signaling molecules and pathways involved in this communication. Discuss the role of the gut microbiota in the Gut-Brain Axis, including the specific mechanisms through which the gut microbiota influences the communication between the gut and the brain.

When discussing the effects of stress on intestinal permeability, consider providing more information on the potential consequences of increased intestinal permeability and how it may impact overall health and well-being.

Response: Thank you for the critique. We included a segment on mechanisms, largely supported by alteration of tryptophan metabolism by the gut microbiota. The section on intestinal permeability was also expanded. Several supporting references were added.

Critique:

  1. Gut Microbiome and its role in the Gut-Brain Axis: When discussing the influence of gut microbiota on neurological and mental disorders, provide a broader overview of the research findings in this area, including the potential mechanisms through which gut microbiota may contribute to these disorders. Explore the potential therapeutic implications of manipulating the gut microbiome to improve neurological and mental health, including the use of prebiotics, probiotics, and fecal microbiota transplantation.

Response: Thank you for the critique. We expanded several sections as per reviewer critique and supporting references have been added.

Critique:

  1. Short Bowel Syndrome and Gut Microbiome: When discussing the role of antibiotics in SBS patients, consider elaborating on the potential benefits and drawbacks of antibiotic treatment, including the impact on gut microbiome composition and the risk of antibiotic resistance. In the comparison of SBS and inflammatory bowel disease (IBD), provide a more detailed overview of the similarities and differences in gut microbiome composition and discuss the potential implications for the pathogenesis and treatment of these conditions. When discussing the impact of anatomical location and length of resection on the gut microbiome and gut-brain axis in SBS patients, consider elaborating on the potential mechanisms underlying these associations and their clinical significance.

Response: Thank you for the critique. We have added information on antibiotic usage.

The extent of bowel resection for SBS and IBD were expanded. More details on IBD have been added, keeping the focus of this manuscript on SBS. References have been added supporting the writeup.

Critique:

  1. Therapeutic Approaches to the Microbiome: Expand your discussion of probiotics to include a wider range of potential benefits and risks in the context of SBS. Discuss the various factors that could influence the effectiveness of probiotics, such as strain selection, dosage, and duration of treatment. Consider discussing the use of prebiotics as an alternative or complementary approach to modulating the gut microbiome in SBS patients. Explain the potential benefits and drawbacks of this approach compared to probiotics. Address the potential role of fecal microbiota transplantation (FMT) in restoring gut microbiome alterations in SBS patients. Discuss the available evidence, potential risks, and future research directions for this approach. Discuss dietary interventions that could be employed in conjunction with microbiome-targeted therapies to improve outcomes in SBS patients. Consider the potential benefits and challenges associated with different dietary strategies, such as specific carbohydrate diets or fiber supplementation.

Response: Thank you for the critique. We have added information on FMT in SBS. We have added verbiage to the probiotic usage as well as potential endotoxin mediated injury mechanisms. New information on diet and SBS has been included, aligning with the scope of this review. Supporting references have been included.

Critique:

  1. Conclusions: Emphasize the importance of future research in uncovering the complex relationships between the gut microbiome, the gut-brain axis, and SBS. Highlight the potential for personalized medicine approaches based on individual gut microbiome profiles.

Response: Thank you for the critique. As per reviewer critique such has been added to the manuscript.

Round 2

Reviewer 2 Report

Please fix the formatting of the bibliography and text.